# Analysis of the Impact of Acoustic Vibrations on the Laser Beam Remelting Process

**DOI:** 10.3390/ma15186402

**Published:** 2022-09-15

**Authors:** Arkadiusz Krajewski, Paweł Kołodziejczak

**Affiliations:** Faculty of Mechanical and Industrial Engineering, Warsaw University of Technology, 85 Narbutta Str., 02-524 Warsaw, Poland

**Keywords:** laser beam welding, mechanical acoustic vibrations, structure properties, hardness, chemical composition

## Abstract

The article contains an analysis of selected aspects of the structure and properties of laser remelting of low carbon steel supported by acoustic vibrations obtained in the research presented in (A Novel Method of Supporting the Laser Welding Process with Mechanical Acoustic Vibrations). Due to the assumptions made in this publication, it was necessary to deepen the analysis of the obtained results. The correlation of such factors on the structure as the frequency of vibrations (50, 100, 1385 Hz), their propagation through the short-lived liquid phase and changes in the structure, chemical composition, and hardness of the characteristic zones of the obtained remelting was considered. The remelting obtained with the participation of resonant acoustic vibrations with a frequency of 1385 Hz was subjected to thermo-mechanical analysis. A characteristic “bandwidth” pattern was revealed in the structure. In the present article, a thermo-mechanical analysis of the cause of its formation was carried out by comparing it with the remelting obtained at lower frequencies. As a result of the analysis, it was found that the band structure was characterized by 7 to 8 areas up to approximately a 90 µm depth, which showed dark and light zones. These areas differed in carbon content, hardness, and width. The analysis of vibration propagation helped to determine that in the time of crystallization of the molten metal pool, the transition of the vibration wave lasted through 7–8 minima and maxima. This fact allows us to assume with a high probability that it is the result of the applied resonance frequency.

## 1. Introduction

The assistance with mechanical vibrations of many manufacturing processes is relatively widely used in the industry in many fields. There are more and more trials, experiments, and even the use of mechanical vibrations in bonding processes. Most often, in the literature, researchers use of mechanical vibrations was introduced by physical contact with the workpiece or welded parent metal [1,2,3,4]. In item [1], the influence of ultrasonic vibration energy on the welding of AISI 321 stainless steel was considered. Researchers found that ultrasonic vibrations increased the concentration of imperfections in the weld, but at the same time, when the vibration power reached 600 W, an increase in tensile strength was observed compared to the native material. In item [2], Al alloy was welded using the MIG method. The native material was also a waveguide. Investigations of the effect of ultrasonic vibrations on the liquid metal pool from Al alloy by the MIG method were carried out. It was observed that ultrasonic vibrations changed the weld structure depending on the wave phase. Ultrasonic vibrations with a frequency of 20 kHz were also introduced during laser welding of Hastelloy C-276 and 304 austenitic steel [3]. It was found that cavitation, which was the effect of vibrations, intensified convection, and diffusion of elements. This led to a homogenization of the structure. In item [4], laser welding of various butt joints made of low-alloy steel was carried out. The influence of the direction of vibration introduced on the microstructure was investigated. It turned out that vibrations with a frequency of 18 kHz reduced the risk of cracking and improved the homogeneity of the structure. The influence of ultrasonic vibrations on the morphology, microstructure, and properties of the laser-welded 304 steel joint was analyzed [5]. As a result of the introduction of vibrations, it was possible to reduce the width of the unmixed zone in the weld. The uniformity of the elements distribution improved, and the average grain size decreased from 1.03 μm to 0.55 μm. The authors [5] put forward the thesis that the acoustic pressure in the liquid pool caused grain fragmentation. Moreover, the vibrations increased the diffusion coefficient and the cooling rate, which contributed to the inhibition of element segregation. In this way, the resistance of the weld metal to intergranular corrosion was improved. The load capacity of the joint increased by 12.6%. An increase in the microhardness of the weld metal and a reduction in the grain size were observed. In [6], ultrasonic vibrations were used during welding of aluminum alloys with the Laser-MIG hybrid method. A reduction in porosity and the conversion of columnar grains into equiaxial grains were observed. Moreover, under the pressure of 132 N generated by the vibration system, the penetration depth increased from 6.6 mm to 4.2 mm. This change was explained by the dispersion of vibrations in the vicinity of the arc plasma in the steam channel of the laser beam. At a pressure of 132 N, the porosity decreased from 5.66% to 1.05%. With increasing vibration energy, the width of the columnar grain zone decreased due to cavitation. However, the authors [6] do not describe the technical method of introducing the vibrations to the surface of the parent material. An interesting method of introducing ultrasonic vibrations to the welded aluminum alloy plate was used by the researchers [7]. Eight ultrasound heads were rigidly fixed with special clamps in such a way that they adhered almost over the entire surface of the sheet. It turned out that the ultrasonic vibrations caused cavitation, which reduced the grain size by approximately 50%. An extremely unique process of explosive welding of dissimilar plates was followed by the implementation of ultrasonic vibrations during bonding [8]. It was shown that as a result of the application of vibrations, the strength and hardness increased as well as the reduction of the liquid fraction in relation to the vibration-free explosion welding. In the work [9], SUS 301 steel was laser welded with the participation of ultrasonic vibrations. The vibrations were introduced through a coupling agent in contact with the parent material. The advantage of this method was the ease of adaptation to different sample geometries. The disadvantages were the necessity to use a coupling medium and a large loss of vibration intensity as well as technical difficulties with the configuration of the setup. The authors [10] use mechanical vibrations with a frequency of 522 Hz and 1331 Hz during laser welding of Inconel 718. The lowest average hardness (331 HV) in the weld was obtained for the introduction of vibrations with a frequency of 522 Hz. The use of vibrations with a frequency of 1331 Hz reduced the number of Laves phases, and their morphology was in the form of granules and stripes. It is worth mentioning one more work in which the effect of vibration on microstructure and fatigue properties of 6082 CMT-welded joints were researched [11]. Low-frequency vibrations were applied, which degassed the weld and reduced the residual stresses by approximately 23 MPa. Increasing the frequency of vibrations to 1222 Hz made it possible to achieve 84% of the strength of the parent material.

Mechanical vibrations introduced in such a direct way require a constant and controlled contact with the surface of the processed elements, which makes it difficult to ensure the appropriate amplitude and constant phase shift at the welding pool. In industrial conditions, it is extremely important to ensure constant and repeatable parameters characterizing the welding process and the introduction of mechanical vibrations. For this reason, it is worth considering the possibility of using methods of introducing vibrations without the need of physical contact. Researchers in [12] used electromagnetic wire wobble during a laser welding process with 316L stainless steel. The use of wobbling wire allowed for the enlargement of the pool of molten metal and improved wettability of the side walls of the weld groove. As a result of the wire wobbling, the face of the weld was concave, which made it easier to arrange the next weld bead. A very similar approach is presented in [13]. The use of vibrations induced in the TIG process helped to obtain a structure with an average grain size of 39 µm. Ultrasonic vibrations with a power of 0.9 kW and a welding current of 65 A increased the maximum bond strength of ultrasonically treated galvanized steel/Mg alloy by 14.6%. Another interesting case of contactless introduction of vibrations is presented in [14]. The authors welded with the GMAW method. Ultrasonic vibrations were introduced parallel to the axis of the torch. This led to an increase in the volume of the molten metal pool by 49% and an increased penetration depth of the Al alloy weld. Some interesting ideas for introducing ultrasonic vibrations during the laser welding of aluminum alloy EN AW-6082 were proposed in the article [15]. The vibrations were introduced through the fluid as a coupling factor with the native material. This solution has advantages and disadvantages. On the one hand, it allows easy adaptation to the geometry and dimensions of the parent material, and on the other hand it requires a complex design of the setup. One of the methods of generating ultrasonic vibrations during welding processes is pulsation of the laser beam [16]. In this work, researchers scanned the surface of amorphous molybdenum disulfide with a laser beam with a wavelength of 355 nm, frequency 35 kHz, and pulse width 6 ns. The heating beam was equidistantly shifted by approximately 50 µm with respect to the remelting beam. The pulsating effect of the laser beam caused thermoplastic deformations in the processed film. The applied system of two beams allowed for a 10% reduction in the power required to initiate the formation of crystals. Another example of generating ultrasonic vibrations “in situ” during the welding process of CALM steel is presented in [17]. The work uses the TIG method, modulating the arc with vibrations at a frequency of 50 kHz. The application of this solution allowed for an even distribution of carbides in the remelted zone and improved the mechanical properties of the weld metal and HAZ. The increase in the vibration strength initially causes an 80% increase in the toughness of the weld metal and then begins to decrease. The hardness and tensile strength of the weld metal decreased slightly. The original idea of introducing vibrations into the wire, fed during GMAW welding of Al-Cu alloy, is studied in [18]. During GMAW welding with the participation of ultrasonic vibrations, it was found that the penetration and width of the weld had increased. As a result of the change in the distance of the vibrating system from the welding site, the way in which the metal passes through the electric arc has also changed. The mixed traversal of the metal appeared when the distance was 20–22 mm. Improvement in the structure of the welds was achieved as a result of cavitation. A new possibility of introducing vibrations by shot blasting during MAG welding of P235GH low-carbon steel was presented in [19]. Generating vibrations in this case in the native material does not require physical contact with any vibrating system. The vibrations are introduced by the shot having a momentum that hits the surface of the parent material. In a way, it is a non-contact method of transmitting vibration energy. It was observed that in the area of the face the grain size was reduced, and the islets of perlite disappeared. In the heat affected zone, a breakdown of dendrites was observed, which produced more equiaxial grains of smaller size. The hardness distribution in the shot blasted joint shows flattening, and the average hardness value is lower by approximately 15 to 20 HV0.1. The proposed method of introducing vibrations by shot blasting is generally common in the industry. In a work [19], it was shown that it is possible to obtain beneficial effects when using this method. In addition, it does not require large investments and is not particularly sensitive to changing processing conditions. Ultrasonic mechanical vibrations can also successfully affect the underwater welding zone [20] in underwater welding (FCAW). It was observed that the vibrations transmitted through the water environment influence the behavior of the molten metal pool, the structure, and the strength of welded joints. When using ultrasonic vibrations in underwater welding, grain fragmentation and a reduction in their size were observed. The amount of polygonal ferrite (PF) decreased. The grain size of the boundary ferrite (GBF) reduced from 34 to 10 µm and the hardness increased from 204 to 276 HV. There was also an increase in the tensile strength of the joint from 545 to 610 MPa and an increase in impact strength from 65 to 71 J/mm^2^. Ultrasonic vibrations used for the transverse vibration of the wire filler metal during TIG welding of ferritic steel were the subject of research in [21]. The conducted research showed that the application of ultrasonic vibrations and increasing the welding speed produced more equiaxial grains in the structure. The use of ultrasonic vibrations increased the tensile strength of the weld and the elongation value. So far, no attempts have been made to introduce acoustic vibrations through the gas medium, apart from work [22]. This article discusses preliminary attempts to use this method in the process of laser welding of low-carbon steel and to introduce acoustic vibrations. Both the introduction of vibrations, which require physical contact with the parent material near the heat source, and the non-contact can cause significant changes in the structure and properties of joints or coatings. However, methods of introducing mechanical vibrations without contact with hot parent materials during welding have the advantage of being able to provide consistent and repeatable conditions around the molten metal pool without exposure to high temperatures. Therefore, it is worth making efforts to investigate various methods of contactless introduction of vibrations in the vicinity of the welded zone. Various methods of contactless introduction of vibrations are discussed in the article introducing vibrations through the wire as a filler metal, the use of oscillation of the electric arc in the TIG and MIG methods, and pulsations of the laser beam. In the work [22], an interesting idea was proposed to introduce vibrations through a gas environment into the molten metal pool during laser beam remelting of P235GH low carbon steel. Acoustic vibrations with a frequency of 1385, 110, and 50 Hz were introduced into the pipe along the axis and transversely from the outer surface. There were presented the results of structure studies after remelting with a CO_2_ laser, hardness distributions and analysis of the weld shape of the laser impact areas. The formation of the “bandwidth” structure in the obtained remelts, which changed due to frequency of applied acoustic vibrations, was subjected to a deeper analysis and its results are presented in this article. The results described in the article showed that there is a technical possibility of effective influence by contactless introduction of acoustic vibrations on the structure and properties of welds with the participation of a CO_2_ laser. The comparative analysis showed that there are differences in the achieved effects, depending on the frequency of acoustic vibrations and the direction and place of their introduction. At the same time, it is worth noting that while completing the results of the metallographic tests, significant differences in the shape of the obtained remelts existed. Both the shape factor and the hardness distributions significantly depend on the introduced acoustic vibrations. The results of the analysis carried out are therefore a continuation of the study [22] and constitute a comprehensive methodological and substantive whole with this work.

## 2. Materials and Methods

In the experimental tests, the previously obtained results [22] were used. Weld tests were carried out on a pipe made of P235GH steel (acc. EN 10204), with a diameter of 60 mm, a wall thickness of 3.2 mm, and a length of 500 mm. The chemical composition of this steel contained: 0.11% C, 0.42% Mn, and 0.016% Si. A CO_2_ laser of the VFA2500 type from Wegmann-Baasel (Aschheim, Germany) was used for remelting. It can generate pulses or operate continuously with a power of 100 to 2500 W. The wavelength of the laser light was 10.6 µm, the diameter of the laser beam on the outer surface of the material was 1 mm, the focus was 0.4 mm below the surface, beam mode was TEM 10, and beam power 1150 W. Remelting was carried out under the mixing of CO_2_, N_2_, and He gases. Shielded gases were supplied at a capacity of 10 L/min. The welding speed was 0.0083 m/s (0.5 m/min). Acoustic vibrations were introduced by means of a setup consisting of a harmonic generator 1023 Bruel and Kjaer type with an amplifier 2734 (Bruel and Kjaer, Naerum, Denmark) and Bruel and Kjaer sound sources type 4295. The acoustic pressure was monitored with a Bruel and Kjaer type 2236 m. The aim of this study is to try to find the cause that led to the formation of the “bandwidth” structure in remelts, especially in the case of using vibrations with the resonance frequency 1385 Hz of the parent material.

To achieve this goal, it is necessary to analyze the phenomena which occurred during the propagation of acoustic vibrations in the environment, as well as in the tube melted with a laser, the heat flow and the pressure or stresses in in the steel. Figure 1 measures all important elements of the experiment.

While the heat flow during welding can be treated as a constant and continuous phenomenon, the introduction of mechanical vibrations to the base material cannot. The nature and form of the vibrations, the introduced sound energy/pressure, and the direction and environmental conditions inside and outside the pipe are important here. To interpret the effects of thermal phenomena, solutions based on Rykalin’s studies [23,24] were used. In the analysis of mechanical vibrations, the occurrence of several different effects was assumed: the propagation of acoustic vibrations in the air and the standing wave of mechanical vibrations in the base material.

Therefore, the residence time of the metal in the liquid state *t* [s]:(1)$$t=\frac{Q}{2\pi kvT_{L}}$$
where *Q* = 1150—thermal power [J/s], *k* = 58—thermal conductivity coefficient [J/msK], *v* = 0.0083—heat source speed [m/s], *T_L_* = 1440—melting point [°C]

Considering declared values of the variables into the Equation (1), the result time value is *t* = 0.00242 s and for this value the remaining calculations describing the course of vibrations and stresses in the native material of the pipe will be performed.

The acoustic pressure *P* [Pa] in the air, due to vibrations of a given frequency, is described by the relationship [25,26]:(2)P=2 π ρ c f A
where *ρ*, *c*, *f*, *A* are the density of the medium [kg/m^3^], the speed of propagation of the acoustic vibration wave [m/s], the vibration frequency [1/s], and the vibration amplitude [m], respectively.

The propagation of acoustic vibrations in the air is described by the relationship [27]:(3)Acos(2πft−2t)+Acos(2πft+2t)
where *A*—vibration amplitude [m], *f*—vibration frequency [Hz], *t*—time [s].

Acoustic vibrations causing air pulsation cause also longitudinal stresses *σ* and transverse stresses *τ* in steel, and these stresses can be easily converted into reduced stresses *σ_Z_* according to the Huber hypothesis [28].
(4)$$\sigma =P\left[\cos\left(2\pi f\frac{\lambda }{c}+\varphi \right)+\cos\left(2\pi f\frac{\lambda }{c}-\varphi \right)\right]$$
(5)τ=Pν[cos(2πf λ/c+φ)+cos(2πf λ/c−φ)]
(6)$$\sigma_{Z}=\sqrt{\frac{\sigma^{2}}{2}+3\tau^{2}}$$
where *P*—sound pressure [Pa], *φ*—phase shift [rad], *ν*—Poisson’s ratio [-], *l*—wavelength [m].

The waveform of acoustic vibrations may also be influenced by the divergence of the beam *b*, defined as the distance between its contours at the distance *x*, which can be approximated by the equation [25]:(7)$$b=2ax\frac{\lambda }{D}$$
where *a*—coefficient depending on the pressure drop value (0–1.22) [-], *D*—effective diameter of the vibration transducer [m].

During the remelting experiments, the acoustic pressure was kept constant at 100 dB, which provided an effective pressure value of 2 Pa. This fact made it possible to determine the amplitude of pressure in air and steel for each applied frequency of acoustic vibrations 50, 110, and 1385 Hz (Figures 2, 3, 8, 9, 14, and 15).

Assuming the sound pressure of the air generates stresses in the material of the steel tube, the analysis of the waveform of vibrations in the steel was conducted for the applied frequencies of vibrations propagating in the determined duration of the liquid phase (Figures 4, 10, and 16). The course of pressure changes in the steel and should result in the stresses described by Equations (4)–(6) and illustrated by Figures 7, 13, and 17.

The analysis considered the divergence of the beam in the air and in the steel because both the waves reflected from the inside of the pipe and those propagating in the steel may cause the interference phenomenon. The results of this analysis are illustrated in Figures 5, 6, 11, 12, 18, and 19. The data and parameters necessary for the analysis are presented in Table 1.

After the analytical procedure was performed, structural studies were conducted using an OLYMPUS BX51M optical microscope (Olympus Optical Co., Ltd., Tokyo, Japan) with Stream Essential v. 2.3 software (Olympus Optical Co., Ltd., Tokyo, Japan), an extended microscopy JEOL electron tube type. ICM 7000-4E and the supplementary Vickers hardness measurements at 100 g were obtained on a LEITZ MINILOAD 8375 device. Hardness is summarized as the average of five measurements from a representative area.

The acoustic pressure in the air of 2 Pa helps to determine the amplitude of vibrations from the Formula (2), which is marked in the rectangle in Figure 2 (1.575 × 10^−5^ m). Then, this value was transferred to the diagram on Figure 3, which helps to determine the value of the pressure in the steel (2.3 × 10^5^ Pa).

Knowing the crystallization time (0.00242 s) of the molten metal pool calculated from Formula (1), which is constant for all cases due to the constant power of the laser beam and the constant welding speed, the course of the vibration wave in steel can be determined from Formula (3) (Figure 4).

As we can see in Figure 4, less than ¼ of the vibration wavelength passes through the molten metal pool during crystallization. Another parameter that will indicate the possibility of vibration interference is an acoustic beam divergence. Assuming the maximum length of the melted steel pipe as 0.5 m, its maximum value can be determined from the dependence (7). It will be important in the analysis of the beam path inside the pipe during the axial propagation of vibrations (Figure 5).

The divergence of the vibration beam calculated from Formula (7) is 10 times greater in steel than in the air, as shown in Figure 6.

The known value of the acoustic pressure exerted in steel and the wavelength of vibrations (Table 1) with a given frequency (here 50 Hz) helps to determine the distribution of longitudinal stresses *σ*, transverse stresses *τ*, and reduced stresses *σ_Z_* in steel from Formulas (4)–(6). We can see it in Figure 7.

As can be seen in Figure 7, normal, shear and equivalent stresses in steel oscillate on a significant wavelength. In this case, it is 118.8 m (Table 1) and results from the vibration frequency. The pipe length on which the process was carried out was 0.5 m, so a small part of the vibration course was used in this case.

Using Formula (2), the amplitude of vibrations was determined at a constant value of the acoustic pressure of 2 Pa (Figure 8). The value of the vibration amplitude 7 × 10^−6^ m made it possible to determine the pressure in the steel in Figure 9 (2.25 × 10^5^ Pa).

The crystallization time of the molten metal pool (0.00242 s) calculated from Formula (1) allows to determine the course of the vibration wave in steel from the relation (3) (Figure 10).

Figure 10 shows that little more than ¼ of the vibration wavelength passes through the molten metal pool during crystallization. The maximum length of the steel pipe is 0.5 m. The maximum value of divergence can be determined from Equation (7). It helps the analysis of the beam path inside the pipe during the axial propagation of vibrations (Figure 11).

The vibration beam divergence determined in Formula (7) is 10 times greater in steel than in the air. It can be observed in Figure 12.

The calculated value of the acoustic pressure exerted in steel and the wavelength of vibrations (Table 1) with a given frequency (here 110 Hz) help to determine the distribution of longitudinal stresses *σ*, transverse stresses *τ*, and reduced stresses *σ_Z_* in steel from Formulas (4)–(6) (Figure 13).

The normal, shear, and equivalent stresses in steel oscillate on a significant wavelength (Figure 13). In this case, it is 54 m (Table 1) and results from the vibration frequency 110 Hz.

As mentioned earlier, the acoustic pressure of the generated sounds was constant and amounted to 2 Pa. In the case of vibrations with a resonance frequency of 1385 Hz, the amplitude calculated from Equation (2) was (5.77 × 10^−7^ m), which is shown in Figure 14.

The amplitude value determined in this way allowed for the pressure evaluation in the steel (2.25 × 10^4^ Pa). It is shown in Figure 15.

In Figure 16, it is noticed that 7 of the wave minima and maxima pass through the molten metal pool during crystallization. Maximum value of divergence (Figure 17) can be determined from the Formula (7), considering the maximum length of the steel pipe as 0.5 m.

In steel, the divergence of the vibration beam is shown in Figure 18.

The determined value of the acoustic pressure in steel and the wavelength of vibrations (Table 1) with a given frequency (here 1385 Hz) help to determine the distribution of longitudinal stresses *σ*, transverse stresses *τ*, and reduced stresses *σ_Z_* in steel from Formulas (4)–(6). It can be seen it in Figure 19.

The known value of the acoustic pressure exerted in steel and the wavelength of vibrations 4.29 m (Table 1) with a given frequency (here 1385 Hz) help to determinate the distribution of longitudinal stresses *σ*, transverse stresses *τ*, and reduced stresses *σ_Z_* in steel from Formulas (4)–(6). It is shown in Figure 19.

According to the standard [29], for the case of the three-dimensional spread of heat, the cooling time from 800 to 500 °C *t*_8/5_ can be estimated, which determines the critical rate of austenite transformation (8).
(8)$$t_{8/5}=\left(6700-5T_{0}\right)Q\left(\frac{1}{500-T_{0}}-\frac{1}{800-T_{0}}\right)F_{3}$$
where *T*_0_—ambient temperature [°C], *F*_3_ = 1—shape factor for three-dimensional heat flow.

The diagram (Figure 20) shows that the cooling time in the austenite stability area at a temperature below 25 °C does not exceed 1 s. It is not enough to consider phase changes based on the continuous cooling transformation diagram (CCT).

In Table 2 summarizes the determined values of the amplitude, pressure and divergence of the vibration wave.

## 3. Results

To analyze the phenomena that occurred during remelting with the participation of acoustic vibrations, supplementary tests were carried out on the structure, hardness, and chemical composition of a distinctive case. Sample no. 3 was obtained by a laser beam welding with the participation of resonant acoustic vibrations with a predetermined [22] frequency of 1385 Hz. The results of these studies are presented in Figure 21, Figure 22, Figure 23, Figure 24 and Figure 25.

Figure 21a shows the growth of grains in the heat affected zone (HAZ) and some characteristic precipitates that accompany the transition zone (TZ) adjacent to the fused area (WM). The structure in this case does not differ much from that obtained in a conventional welding process. The SEM image (Figure 21b) also does not reveal any spectacular changes. The penetration depth is 200 µm, the width of the face is over 600 µm.

In Figure 22a we can see that there is no significant grain growth in HAZ. Precipitates in the TZ zone still occur, but they are at greater distances from each other. The penetration depth is smaller than in the previous case and amounts to almost 250 µm, the width of the face is slightly less than before (500 µm). Smooth transitions between zones are observed in the SEM image (Figure 22b).

Figure 25a–c presents comprehensive reports on the chemical composition tests in three representative areas, 1, 2 and 6, marked in Figure 24. In band 1, where dark precipitates occur, an increase in carbon concentration was observed. In areas 2 and 6, a decrease in its concentration was noticed. Table 3 shows the bandwidths of the regions and a suggested structure.

In Figure 26a the grain growth in HAZ can be seen. The TZ zone containing dark precipitates occurs over a considerable width of approximately 70 mm. The penetration depth is instantly smaller than in the previous case and amounts to almost 160 µm, but the width of the face streams to 800 µm. In the SEM image (Figure 26b) significant grain size is visible both in the TZ zone and in the WM.

In Figure 27a, no grain growth is observed in HAZ, WM, and TZ. In the TZ zone, the occurrence of dark precipitations is not as clearly visible as in the previous cases. The width of this zone is approximately 50 mm. The penetration depth is over 200 µm, and the width of the face streams to 600 µm. In the SEM image (Figure 27b) the grains size in the HAZ, TZ, and WM are very similar.

## 4. Discussion

The geometry of remelts significantly depends on the frequency of acoustic vibrations used during laser remelting, and this feature was already investigated in previous studies [22]. The analysis of the course of acoustic vibrations in air and in steel was not carried out in the previous article by the authors [22]. Now, the effects of the thermo-mechanical analysis are presented, which were to answer what was the cause of the “bandwidth” structure in the weld metal with the participation of resonant vibrations with a frequency of 1385 Hz. The aim of the authors was also to investigate whether similar structures are formed in the case of other vibration frequencies. Now it is worth emphasizing that based on the results of research and analyses, it has been shown in the current work that the effects of the applied acoustic vibrations during CO_2_ laser remelting depend mainly on their frequency. Vibrations with a resonance frequency of 1385 Hz induce a streaked morphology in the melted area with distinct separations in darker zones. The characteristic band pattern in welds (seven specific areas) is related to the occurrence of seven minima and maxima of vibrations deflection occurring in the estimated short time of crystallization.

In remelts with vibrations assisted at the frequency of 50 and 110 Hz, the areas of the band structure with visible precipitates are not observed. It is true that we cannot see the specific separations, but we can observe alternately lighter and darker areas. The reasons for their formation can be seen in the interference caused by a significant divergence of the wave front, respectively 15 m and 7 m for 50 and 110 Hz. Introduced vibrations with a resonance frequency of 1385 Hz give a beam divergence of 0.05 m, which is smaller than the diameter of the melted pipe and therefore does not affect the remelting structure (Figure 28).

It can be noticed that in the case of using the vibration frequency of 50 Hz with a divergence of approximately 15 m over a distance of 0.5 m, we deal with 11 intersections of wave beams and 12 reflections from the inner wall of the pipe. At a frequency of 110 Hz and a divergence of 7 m over a distance of 0.5 m, there are six such intersections and also six reflections, and at 1385 Hz over a distance of 0.5 m there is not a single intersection and reflection.

Considering the phenomena that occur during laser welding with assisted acoustic vibrations, the greatest effect on the remelting structure is obtained when the resonance frequency is 1385 Hz. On the other hand, the influence of the interference of vibrations inside the pipe on the structure of the obtained welds at low frequencies of 50 and 110 Hz is small.

The study of the chemical composition of the characteristic areas in remelting No. 3 obtained with the support of vibrations with a resonance frequency of 1385 Hz indicate some differences in the concentration of carbon, iron, and manganese. In the place where grayish spot precipitates can be observed (Figure 26), was noted a greater concentration of carbon, approximately 4.34% mass, 16.89% at. While in the areas without such precipitates, the carbon concentration ranged from 2.20% mass to 2.33% mass, 9.45% at. up to 9.66% at. The areas with higher carbon enrichment and those depleted alternately affect the hardness of these zones. The hardness changes cyclically according to the number of minima and maxima of vibrations occurring during the crystallization time. However, no other welding imperfections are observed. It can be concluded that at a constant power of the laser beam and a constant remelting speed, differentiation of the crystallization fronts takes place, which change cyclically under the influence of the resonance frequency of 1385 Hz. In the structure of samples obtained with the participation of other frequencies, no such specific areas of variable chemical composition are observed.

Measurements of microhardness for sample 3 of the area lying at a depth of 85 μm in the zone designated in Figure 24 show cyclical changes in hardness for lighter areas (without precipitates) from 210 HV0.1 to approximately 300 HV0.1 (in darker visible places with participation). Both the chemical composition tests and the hardness measurements suggest an occurrence of changes in the structure and properties of the tested zones, which were formed because of the impact of mechanical vibrations with a resonance frequency. Similar changes of a comparable scale in the remelts cases with the different frequency vibrations are not observed.

## 5. Conclusions

The conducted research and their analysis show that it is possible to effectively influence both the structure of the crystallizing liquid phase and the mechanical properties, e.g., hardness. It has been shown that welds obtained with the participation of mechanical vibrations are fundamentally different from those obtained without vibrations;Moreover, which is very important from the implementation point of view, it has been shown that the contactless introduction of acoustic frequency vibrations effectively affects the properties and structure of the liquid phase welded with the laser beam. This is particularly evident in case of welding, with acoustic vibrations at a resonant frequency. In this case, a specific repetitive morphological pattern was obtained with highlighted areas of variable structure, chemical composition, and hardness. The number of noticed changes coincides with the number of minima and maxima of the wave propagating acoustic vibrations;The result of the caried out analyzes carried out here shows that it is possible to construct a modified laser head for welding processes involving mechanical vibrations. However, to thoroughly answer the question of how acoustic vibrations affect the welding process, a numerical superposition of thermal phenomena and mechanical vibrations will be needed in the future.

## Figures and Tables

**Figure 1 materials-15-06402-f001:**
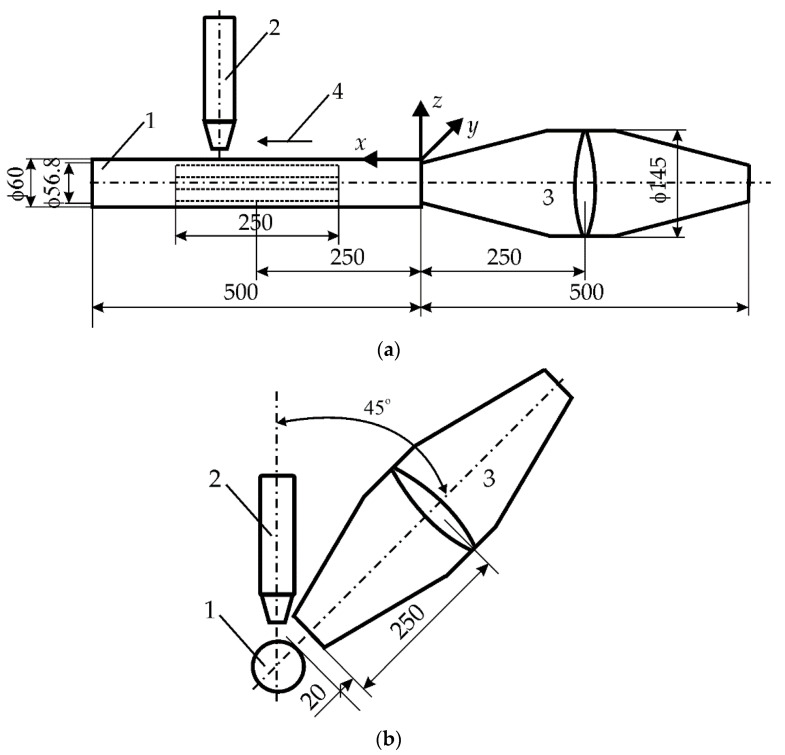
Experimental setup for the laser beam welding and introducing acoustic vibrations: (**a**) setting for the longitudinal introduction of vibrations, (**b**) setting for the transverse introduction of vibrations, 1—steel pipe, 2—CO_2_ laser head, 3—source of acoustic vibrations, 4—direction of moving heat source (all dimensions in mm).

**Figure 2 materials-15-06402-f002:**
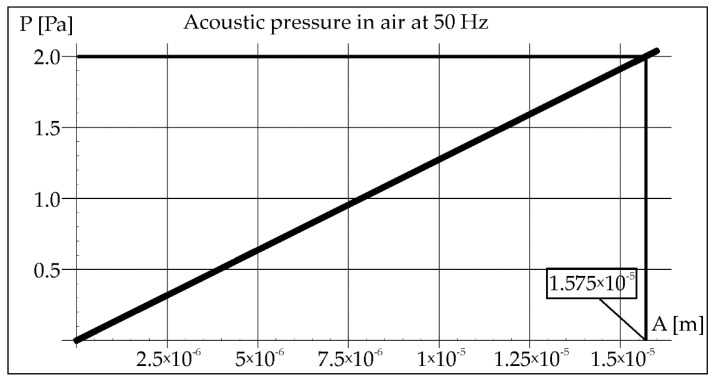
Acoustic pressure in air at frequency 50 Hz.

**Figure 3 materials-15-06402-f003:**
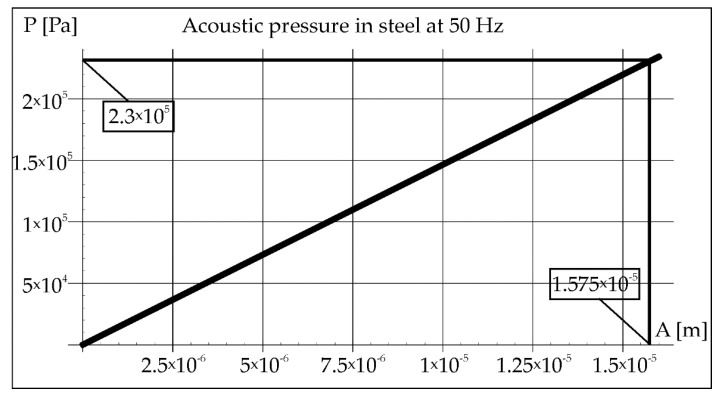
Acoustic pressure in steel at frequency 50 Hz.

**Figure 4 materials-15-06402-f004:**
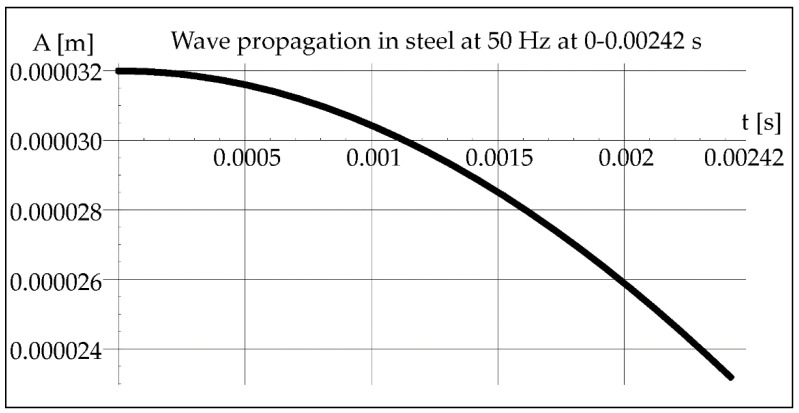
Propagation of acoustic wave in steel at 50 Hz during 0–0.00242 s.

**Figure 5 materials-15-06402-f005:**
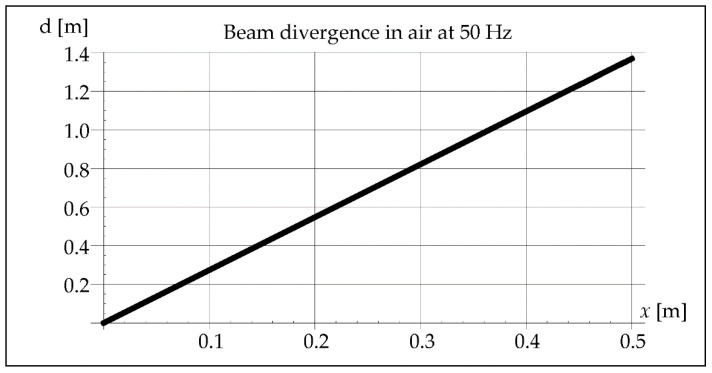
Beam divergence in air at 50 Hz.

**Figure 6 materials-15-06402-f006:**
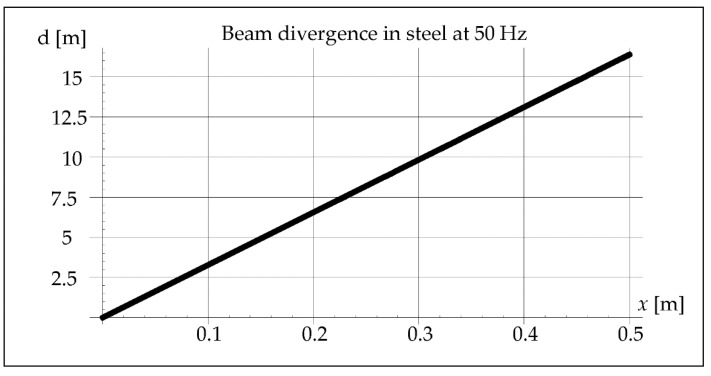
Beam divergence in steel at 50 Hz.

**Figure 7 materials-15-06402-f007:**
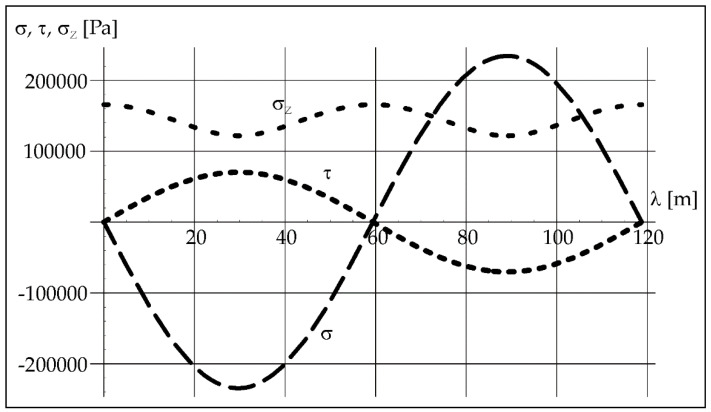
Stress distribution in steel at 50 Hz.

**Figure 8 materials-15-06402-f008:**
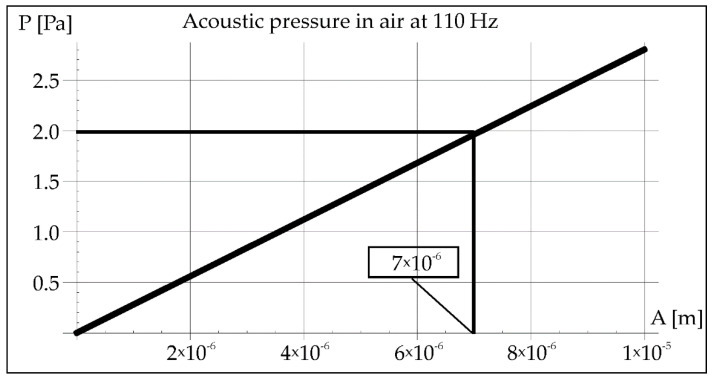
Acoustic pressure in air at resonance frequency 110 Hz.

**Figure 9 materials-15-06402-f009:**
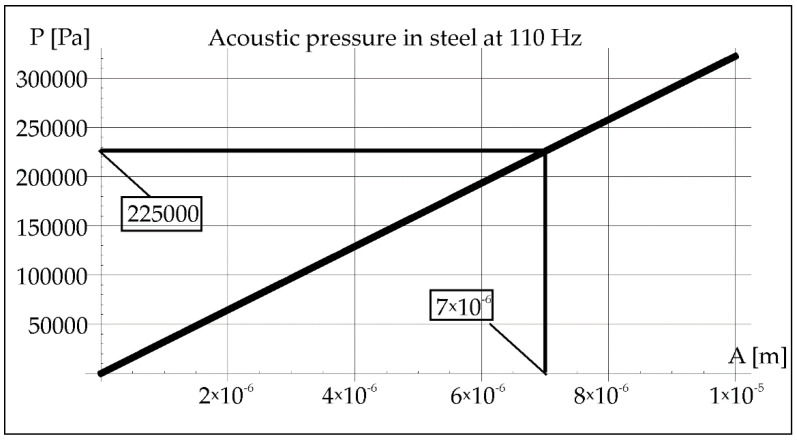
Acoustic pressure in steel at resonance frequency 110 Hz.

**Figure 10 materials-15-06402-f010:**
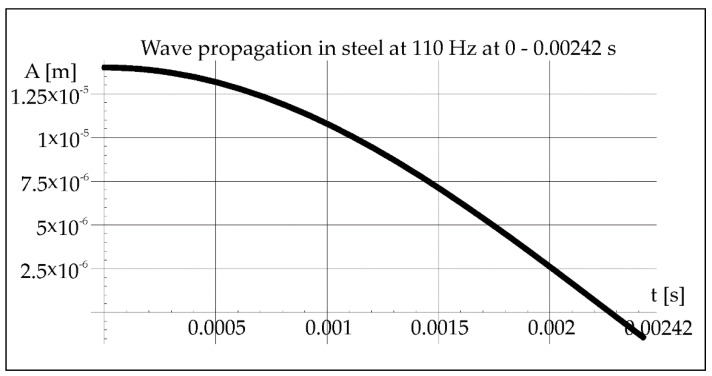
Propagation of acoustic wave in steel at 110 Hz during 0–0.00242 s.

**Figure 11 materials-15-06402-f011:**
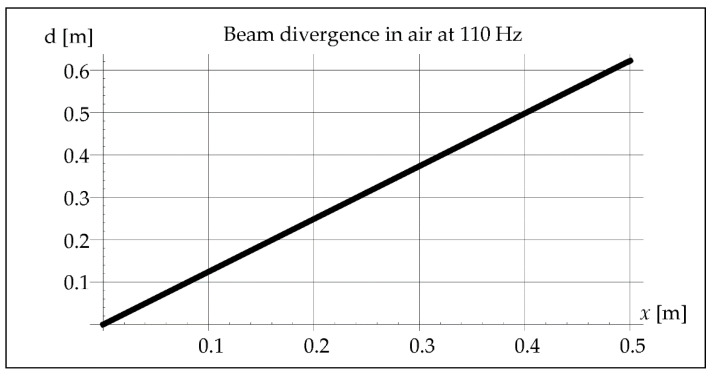
Beam divergence in air at 110 Hz.

**Figure 12 materials-15-06402-f012:**
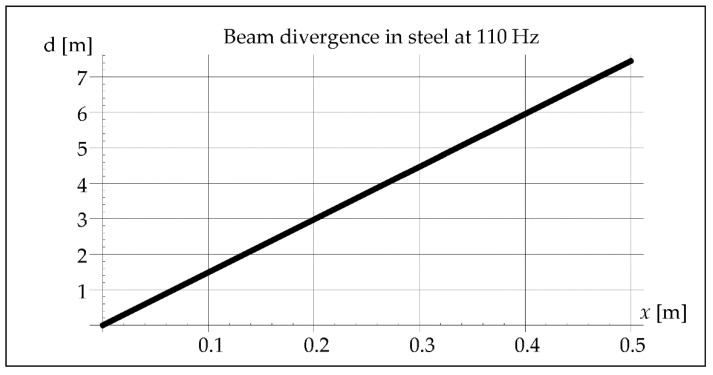
Beam divergence in steel at 110 Hz.

**Figure 13 materials-15-06402-f013:**
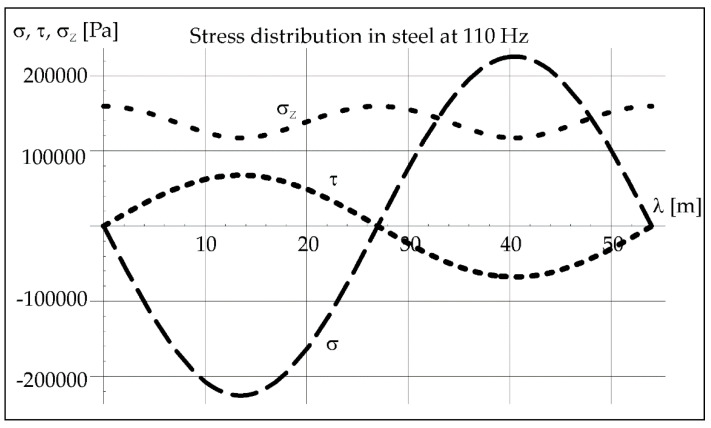
Stress distribution in steel at 110 Hz.

**Figure 14 materials-15-06402-f014:**
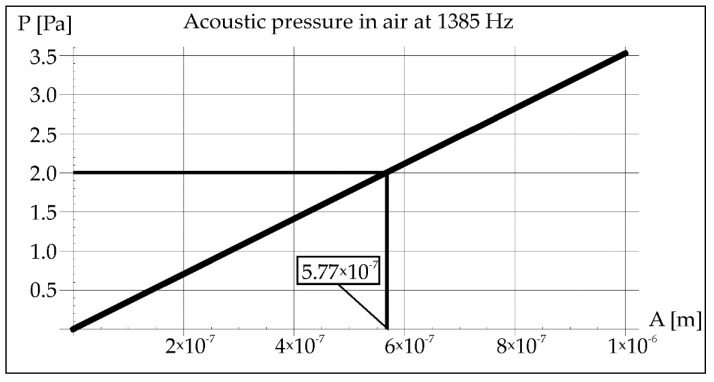
Acoustic pressure in air at resonance frequency 1385 Hz.

**Figure 15 materials-15-06402-f015:**
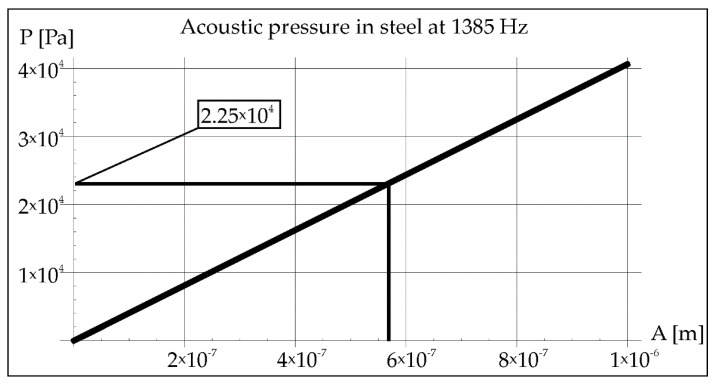
Acoustic pressure in steel at resonance frequency 1385 Hz.

**Figure 16 materials-15-06402-f016:**
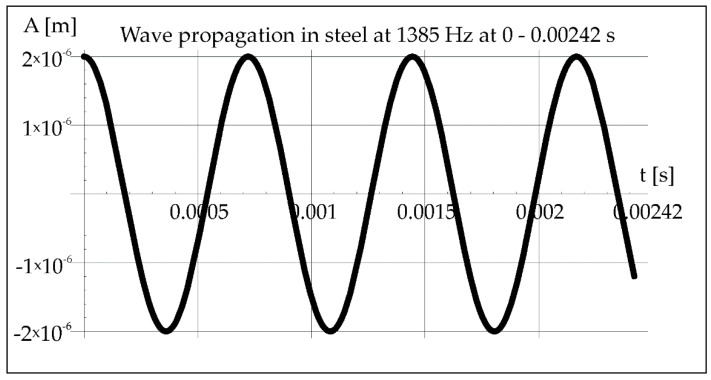
Propagation of acoustic wave in steel at 1385 Hz during 0–0.00242 s.

**Figure 17 materials-15-06402-f017:**
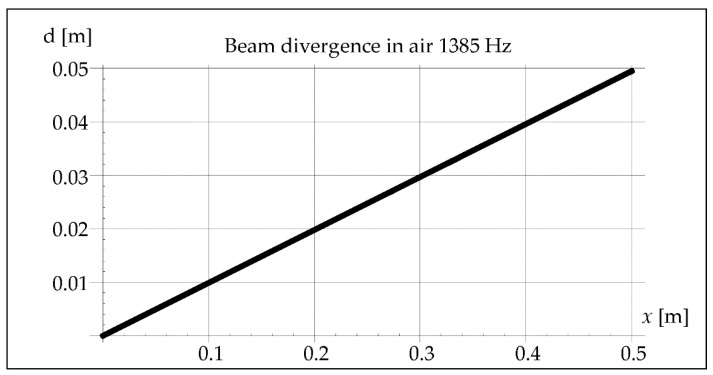
Beam divergence in air at 1385 Hz.

**Figure 18 materials-15-06402-f018:**
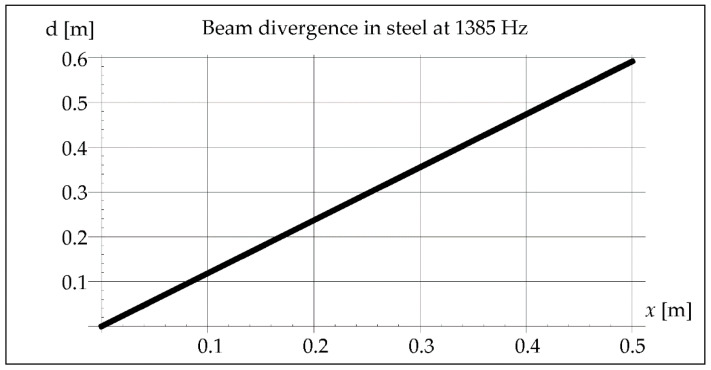
Beam divergence in steel at 1385 Hz.

**Figure 19 materials-15-06402-f019:**
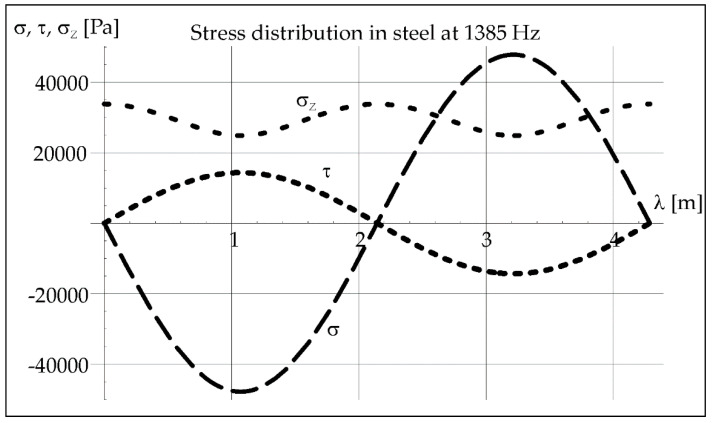
Stress distribution in steel at 1385 Hz.

**Figure 20 materials-15-06402-f020:**
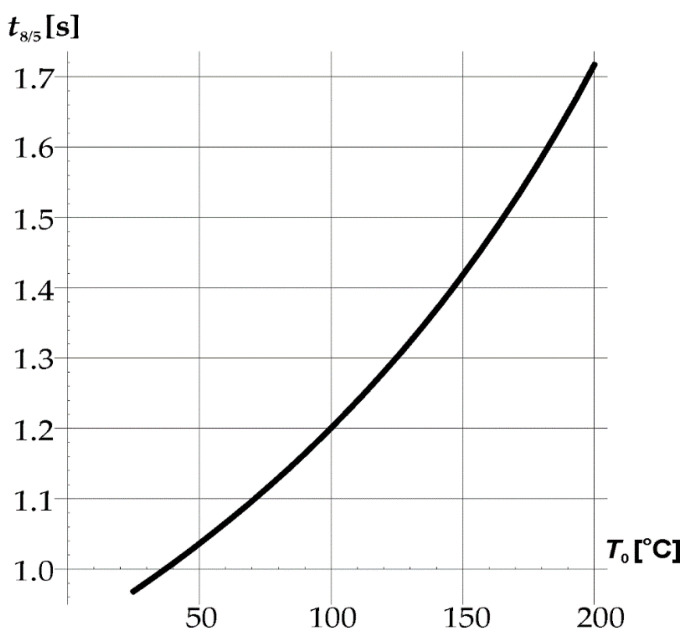
Cooling time *t*_8/5_ depending on the ambient temperature *T*_0._

**Figure 21 materials-15-06402-f021:**
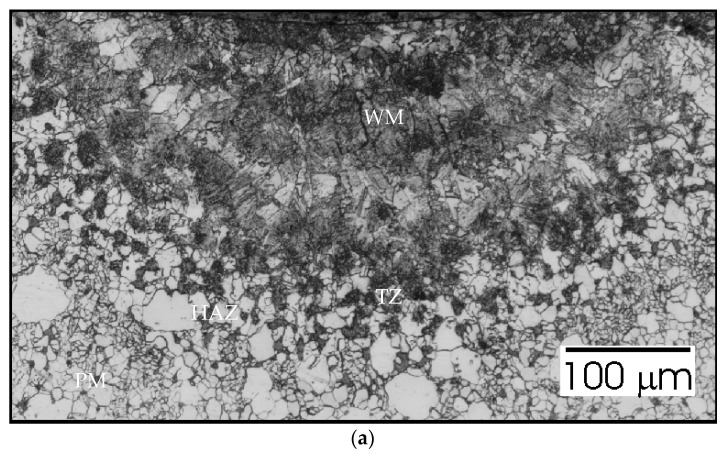
Sample no 1 reached with 50 Hz acoustic vibration delivered transversely: (**a**) Metallography and (**b**) SEM; PM—parent metal, HAZ—heat affected zone, TZ—transition zone, WM—weld metal.

**Figure 22 materials-15-06402-f022:**
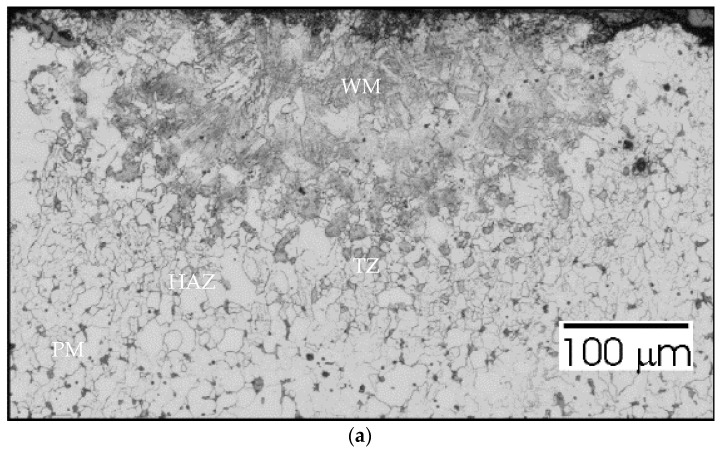
Sample no 2 reached with 50 Hz acoustic vibration delivered parallelly: (**a**) Metallography and (**b**) SEM; PM—parent metal, HAZ—heat affected zone, TZ—transition zone, WM—weld metal.

**Figure 23 materials-15-06402-f023:**
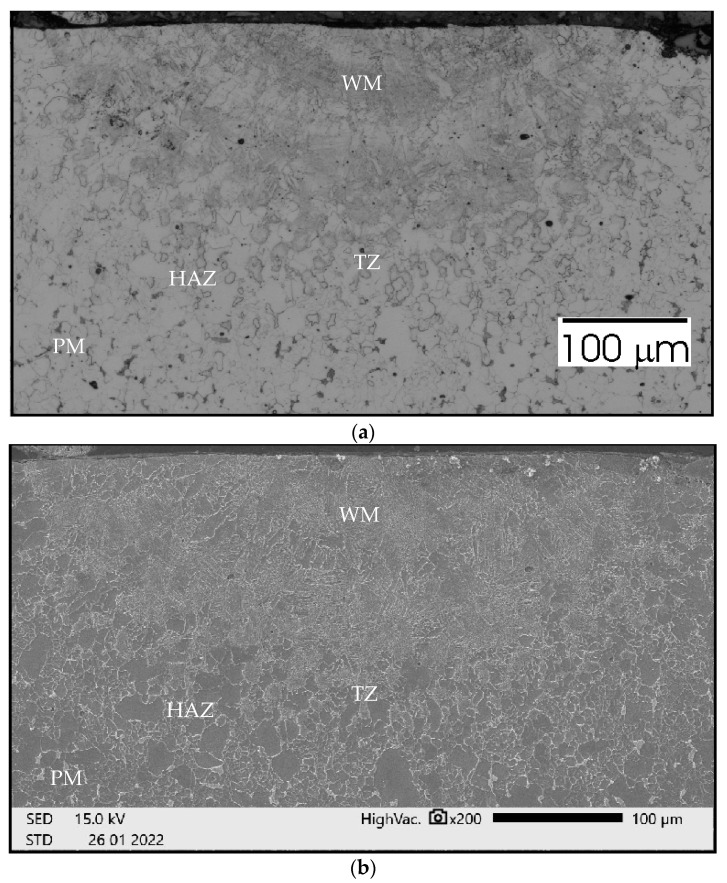
Sample no 3 reached with resonance frequency 1385 Hz acoustic vibration delivered parallelly: (**a**) Metallography (**b**) SEM; PM—parent metal, HAZ—heat affected zone, TZ—transition zone, WM—weld metal.

**Figure 24 materials-15-06402-f024:**
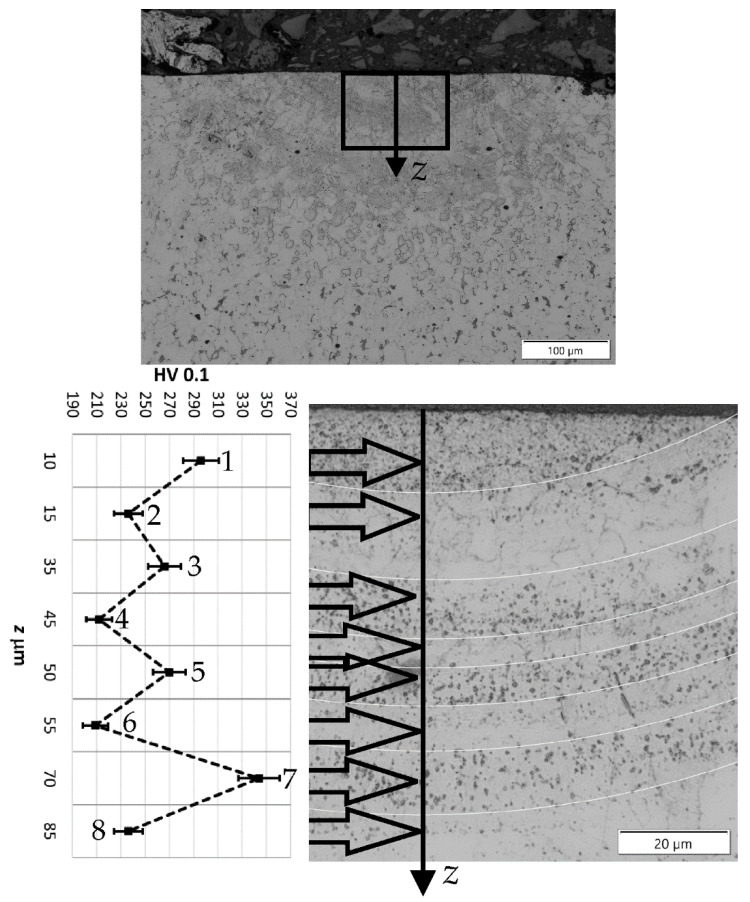
Sample no 3 reached with resonance frequency 1385 Hz acoustic vibration delivered parallelly: hardness distribution; the arrows indicate the area of the hardness measurement.

**Figure 25 materials-15-06402-f025:**
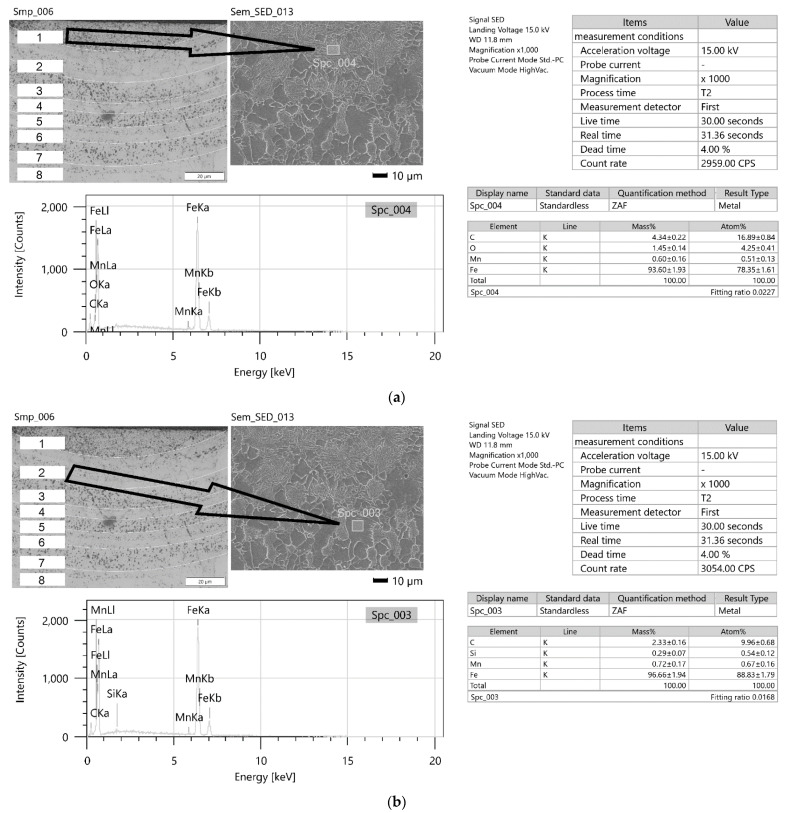
(**a**). Comprehensive report on the analysis of chemical composition for sample 3 from area 1 (Figure 24). (**b**). Comprehensive report on the analysis of chemical composition for sample 3 from area 2 (Figure 24). (**c**). Comprehensive report on the analysis of the chemical composition for sample no. 3 from area 6 (Figure 24).

**Figure 26 materials-15-06402-f026:**
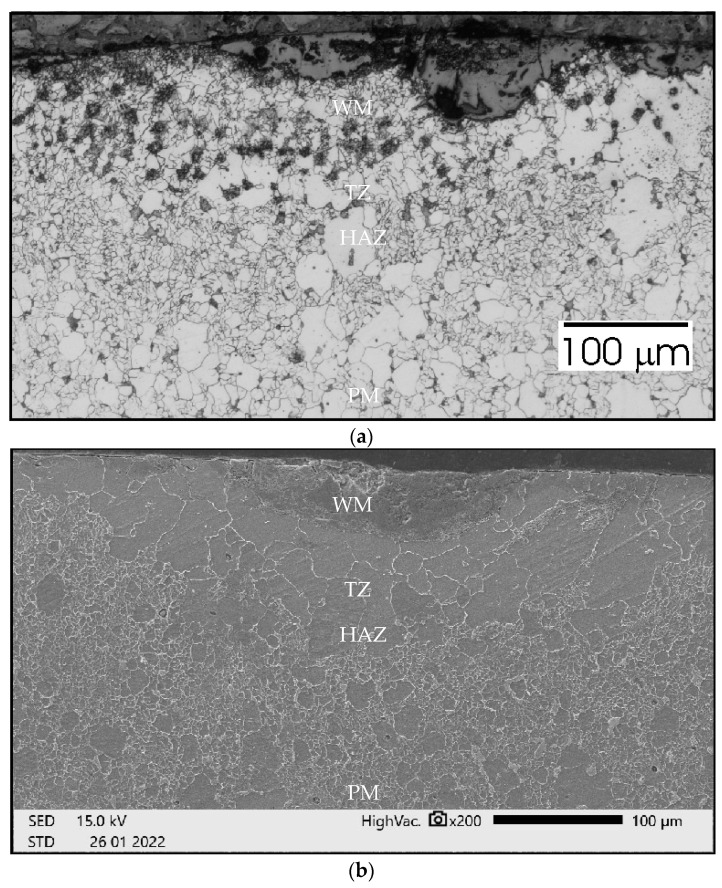
Sample no 4 reached without acoustic vibration: (**a**) Metallography and (**b**) SEM; PM—parent metal, HAZ—heat affected zone, TZ—transition zone, WM—weld metal.

**Figure 27 materials-15-06402-f027:**
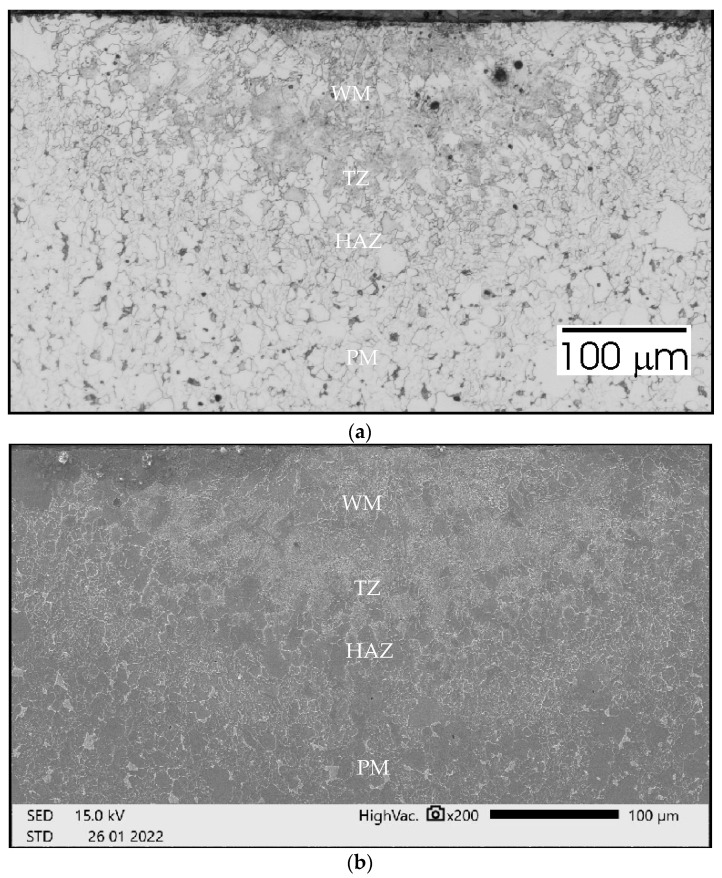
Sample no 5 reached with 110 Hz acoustic vibration delivered parallelly: (**a**) Metallography and (**b**) SEM; PM—parent metal, HAZ—heat affected zone, TZ—transition zone, WM—weld metal.

**Figure 28 materials-15-06402-f028:**
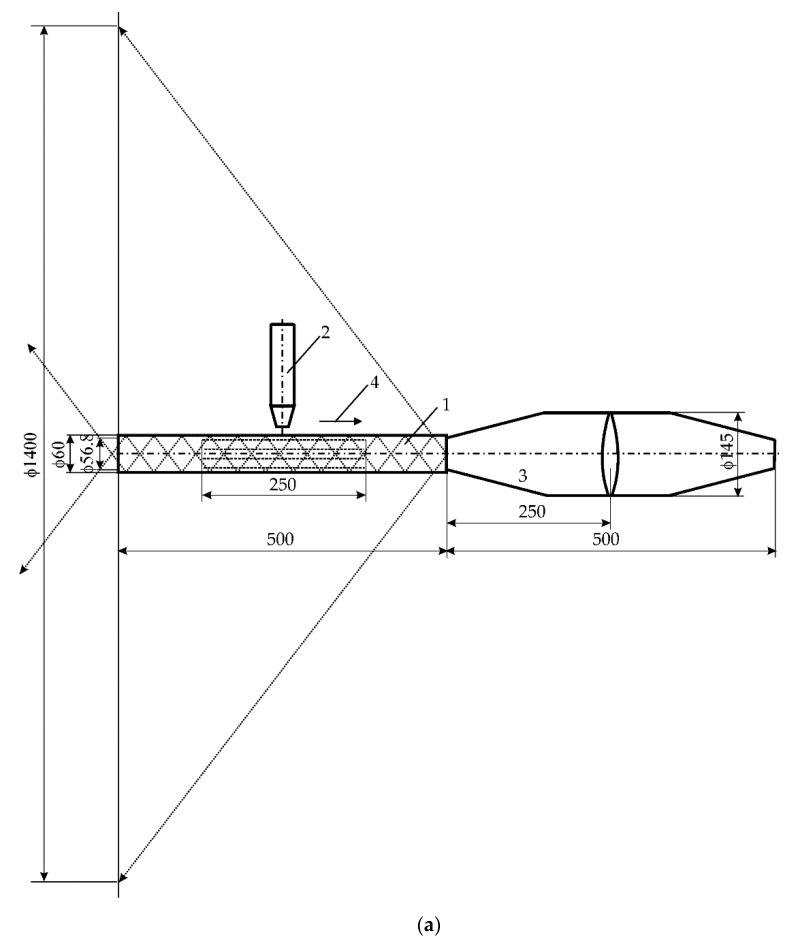
The beam of vibrations in the air inside the pipe: (**a**) vibrations with a frequency of 50 Hz, (**b**) vibrations with a frequency of 110 Hz, (**c**) vibrations with a frequency of 1385 Hz (all dimensions in mm).

**Table 1 materials-15-06402-t001:** Material and characteristic parameters of vibrations supporting laser welding.

Weld No	Frequency *f* [Hz]	Propagation	Wavelength [27] *λ* [m]	Velocity [27] *c* [m/s]	Continuum
-	50	II/⊥	6.62	331	air ^1^
1	50	⊥	65	3250	steel ^2^
2	50	II	118.8	5940	steel ^2^
-	110	II/⊥	3.01	331	air ^1^
5	110	II	54	5940	steel ^2^
5	110	⊥	29.5	3250	steel ^2^
-	1385	II/⊥	0.24	331	air ^1^
3	1385	II	4.29	5940	steel ^2^
3	1385	⊥	2.35	3250	steel ^2^

^1^ density of air at room temperature: *ρ*_A_ = 1.225 [kg/m^3^]. ^2^ density and Young modulus of steel at room temperature: *ρ*_S_ = 7850 kg/m^3^; *E* = 205 × 10^9^ [Pa]. II/⊥—longitudinal/transverse propagation.

**Table 2 materials-15-06402-t002:** Vibration amplitudes in air and steel for a constant sound pressure of 2 Pa.

Frequency *f* [Hz]	Number of Max/Min *	Amplitude [m]	Pressure [Pa]	*σ*z	Divergence [m/0.5 m]	Continuum
50	-	1.575 × 10^−7^	2	-	1.4	air
50	<1	1.575 × 10^−7^	2.3 × 10^5^	12–16 × 10^5^	15.2	steel
110	-	7 × 10^−6^	2	-	7.5	air
110	<1	7 × 10^−6^	2.25 × 10^5^	11–16 × 10^5^	0.62	steel
1385	-	5.77 × 10^−7^	2	-	0.6	air
1385	7–8	5.77 × 10^−7^	2.25 × 10^4^	25–37 × 10^4^	0.05	steel

* Number of max/min—minima and maxima of the propagating acoustic wave.

**Table 3 materials-15-06402-t003:** The width of the bands and the suggested structure in the remelting area of sample no. 3.

*r*_i_ No	Δ [μm]	Structure P/F
1	15	P
2	17	F
3	10	P
4	5	F
5	7	P
6	8	F
7	11	P
8	10	F

P/F—pearlite-like/ferrite-like phase. *r*_i_ No—area No at Figure 23b. Δ—the width of the areas.

## Data Availability

Data sharing is not applicable for this paper.

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
