# Peer review of "Analysis of the Impact of Acoustic Vibrations on the Laser Beam Remelting Process"

_materials, 2022, doi:10.3390/ma15186402_

Round 1

Reviewer 1 Report

The article entitled "Analysis of the impact of acoustic vibrations on the laser beam remelting process" needs lot of improvements in my opinion in order to be considered / accepted to be published in the Materials journal!

The article is linked to the article "A Novel Method of Supporting 11 the La-ser Welding Process with Mechanical Acoustic Vibrations" that was published by 2 of the authors in Materials 2020, but:

1. it is not very clear what this article bring new as compared to the initial article - in the way things are explained in the text! More details / much clear details must be introduced in the Abstract and Introduction section in my opinion. Also in the Introduction section must be introduced new references. It is not at all ok that the most recent references in the paper are dated in 2020 (when previous article has been published) - just 3 references are dated in 2020 / not a single one in 2021 or 2022! Please include in the article at least 30-35 references that are most recent / new / relevant in the field! Please provide more details in the text when specific references are made in the text to the work of other researches. Do not use [5-10] in terms of references, when just one very general sentence it is offered about it.

2. In the Materials and methods section please include more details about the re-melting process - how this was actualy made. It is not enough just to provide evasive reference to [5]. Please provide details about the equipment that was used in the process / parameters that were set / samples that were realized for testing. Please include macro photos of the realized samples accompanied with a table of technological parameters that were used in the manufacturing process and also CAD models of the samples that were subjected for the realized tests. Try to explain in few words from the setup point of view what was different as compared to the article "A Novel Method of Supporting the Laser Welding Process with Mechanical Acoustic Vibrations" in this section again. It is needed - otherwise is hard to comprise exactly what it is new in terms of setup / manufacturing process / parameters - in the current experiments as compared to the initial ones - if there are indeed some differences from this point of view.

3. In terms of Figures / explanations provided in the text about the figures or tables - I didn't personaly appreciated at all how things are provided in the article. For instance, in Line 123 the references aout figures are provided like this. "Figs. 1,2,7,8,13,14)"...in Line 127... (Fig. 3, 9, 15)...in Line 128... Fig. 6, 12, 16....in Line 132...." Fig. 4,5,10,11,17,18"...Please explain each figure separately in the text...It is too evasive in the way things are given...One sentence followed by a reference to 6 or 7 figures like a "salat mixture"...where anyone can understand what he wants in the absence of explanations.

4. I didn't appreciate also that on 6 pages are provided just figures. Please intercalate them with explanations otherwise they are really very hard / impossible to be followed. Please count each figure separately / be consequent with the style of presenting the figures in the text. Don't use please Fig.22.d, Fig.22.e...or Fig.22f so...Please count each figure separately and provide reference to each figure not just some of them...

There are figures that are not at all explained in the text / how so??? There are so many things shown in the figures - how anyone can understand the details that are shown in the figures / the importance of the results achieved - if the details are missing in the text or they are given like Fig.20-24 (only this reference is related to 10 figures or so!!! It is too evasive / too general  and not at all ok how things are provided in the article)...Please take each explanation separately to each figure /please provide details about what is shown in the figures / and please provide details right after presenting each fig...

5. In the discussion section the results are provided, making a reference just on the begining at the reference [5]. Please take each characteristic and provide much clearly on each point which are the extra benefits that this research has brought in comparison with the initial one - "A Novel Method of Supporting the Laser Welding Process with Mechanical Acoustic Vibrations"! I suggest that before going into explanations a table should be included here where things are much clearly emphasized regarding the initial variant and the new proposed variants of the experiments - to emphasize much better the advantages that the new research is bringing to in terms of results. Importance of the results must be discussed in relation with this reference [5], but please include more references to other researches that were made similar in the field by other researchers in the period 2021-2022. Take one look to the intial article "A Novel Method of Supporting the Laser Welding Process with Mechanical Acoustic Vibrations" where at section 4. Discussion there are made at least 10 references to the work of other researchers in the field...This must be considered also here in the case of this article...it is not enough just to make reference to a single paper (published by same authors /by you in 2020).

6. In the conclusions section where it is mentioned that "The result of the analyzes carried out here shows that it is possible to construct a modified laser head for welding processes involving mechanical vibrations. Conceptual work on such a solution is already underway." It sounds very nice, but mysterious and again this conclusion is very evasive...Please provide more details about this "conceptual work" /include a schematic image at least + some explanations that emphasize how from the new conceptual solution look like, why it is considered like this (specific on which particular results) and how this will influence the expected research / results in continuing.

7. There are lot of "spelling and grammar" errors in the text / corrections to be made. For instance the name of the institute of the authors / affiliation is specified that it is different, when actualy it is the same (the authors are coming from one single institution (1), not (1) and (2). There are some word written in this way (Line 13: "co-relation") in the text ("correlation" it is the right form" - This is just an example / there are also many other similar errors in the text. Beside this there are wrong references to some elements provided in the text for some explanations (e.g. see Lines 131 and 132) where it is written like this " The data and parameters necessary for the analysis are presented 132 in Table 1 in [5] and in Table 1 in this article. The resulting values are given in Table 2." In the way things are provided here the aspects are very confusing. References are actually made to 3 Tables (one from other article / one of this article given on page 4 and one given on page 11 of this article! Suggestion is to provide a single Table where all things are being put on the same place, so differences could be much easier to be observed! Of course some comments needs to be introduced in the text immediately after this table will be provided. You cannot comment something on page 4 that is making a reference to a table that will be provided on page 11 in the article, really!!!! Not to be mentioned that you need a listed paper from the previous article where someone can see Table 1 from the initial article to compare it with Table 1 from this article....No way!!! It is really impossible and not very professional to work like that / to follow the progress of the achieved results in this way. Not mentioning that the importance of results provided in Table 2 are not at all explained in the text somewhere...It is just being in a evasive way at Line 133 in the article and that's it...Good luck for the readers...they should be detectives to understand what they want / to detect the differences by putting all 3 tables together! It is really not at all very professional! Also in terms of formulaes...why just [7] of them are counted and the formula presented in Line 185 is not counted as being formula [8]? Please read carefully the article once again and do the necessary corrections - there are some sentences in the text without verbs!

8. There are some details missing in the article, like Author Contributions, Funding, Conflict of Interest (declaration) - please check carefully the template of MDPI Materials journal and make the corrections accordingly!

This being said and taking into consideration the changes that are needed to be done by the authors as they were presented above the decision is for this article (in the form it is now) to be "Reconsidered after major revision"! There are important missing in terms of some experiments / scientific details that must be added in the text as they were requested, but also major changes / revisions are needed in terms of English language and style (the text comprise lot of errors / it is too evasive in many places / scientific soundness and clearness of the article being very very poor as it is provided in this variant. 

The topic is interesting, authors are recommended to take all these observations in positive manner and they must put effort in adding / providing details that were requested / do the corrections as they are needed and finally come up with an updated variant of the article that is expected to be done / delivered in much professional way as it now! 

I encourage the authors to do the changes as they were suggested, so as after the new improved variant of the article will be checked and revised, the article will have the chance to be accepted for being published in the future in MDPI Materials journal! 

Author Response

Thank you very much for critically reviewing our article. All comments and suggestions have been considered in the manuscript text in the appendix. Here are answers to concerns and suggestions.

Ad. 1. After the publication of the article "A Novel Method of Supporting the Laser Welding Process with Mechanical Acoustic Vibrations" in 2020, it turned out that as a result of further research, the "bandwidth" structure of remelt obtained with the participation of acoustic vibrations with a resonance frequency of 1385 Hz was revealed. It was interesting because previous studies using small magnifications did not show it. After finding that such a structure existed, the authors decided to explain why it was so and compare similar results for other cases. Thus, the research program and analysis were structured in such a way as to obtain an answer to what mechanism is responsible for this change and whether it can be observed using other frequencies. Appropriate supplementation was made both in the abstract and in the introduction. In addition, the literature review was supplemented with items from the last two years. Unfortunately, no publication on the use of acoustic vibrations during welding could be found. The only such publication is the above-mentioned publication from 2020. Regarding the state of the art, each item has been discussed separately.

Ad. 2. In the Materials and Methods chapter, the necessary details of the remelting process with the participation of acoustic vibrations and the parameters of the research equipment have been supplemented. Descriptions of the conditions for carrying out structural tests and hardness measurements have been added. As already mentioned, it is explained what is new in the article compared to the previous one.

Ad. 3. Explanations have been added to each figure in the text. The graphs were interpreted, and pictures of the structures were described in the text. The analytical procedure was explained on a case-by-case basis. I believe these changes will be sufficient.

Ad. 4. As already mentioned in answer to note no. 2 and 3 corresponding explanations have been added.

Ad. 5. The analysis of the course of acoustic vibrations in air and in steel was not carried out in the previous article by the authors. Now, the effects of the thermomechanical analysis are presented, which were to answer what was the cause of the “bandwidth” structure in the weld metal with the participation of resonant vibrations with a frequency of 1385 Hz. The aim of the authors was also to investigate whether similar structures are formed in the case of other vibration frequencies. The authors' intentions are given in the introduction. Nevertheless, such an explanation we included in the Discussion chapter.

Ad. 6. I agree that the mere mention of the work on the project does not contain any detailed descriptions. We would not like to disclose them too early. This will probably be the subject of a patent. I can only explain that it will be a specialized laser welding torch, the axis of which coincides with the axis of the laser head.

Ad. 7. Hopefully the bugs and linguistic awkwardness have been removed. We also improved the author affiliation reference. References to the wrong tables have been changed. One of them gives selected vibration parameters and the other shows the obtained results. Therefore, they should not be combined into one table as their content relates to different chapters. Cross-reference to the data in the table of the previous article has been canceled. The numbering of formulas was supplemented so that all appearing in the text have their own unique designation.

Ad. 8. Contribution of copyright, conflict of interest, etc. information has been added as per the MDPI Materials template.

Thanks again for your time and a thorough review of our manuscript. We believe we have managed to make the necessary corrections. We hope that they will prove to be sufficient for the article to be published.

Yours faithfully

Reviewer 2 Report

1. Give colorful pictures in Fig. 1 and Fig. 25 is better.

2. In Table 2. Number of max/min, max what/min what? The details should be indicated in the table.

3. In Figs. 21-24, the label (a)(b)(c)(d) should be put in the top right-hand corner of the pictures.

4. The two pictures in Fig. 22(c) should be separated. And the big arrow in the top half is not necessary, the small arrows in the Lower half have Overlap. Please explain the meaning of the small arrows.Maker the test position and give the hardness value in SEM picture is better to understand.

5. In Figs. 22d-22f, Can the background color of the numbered text box in the figure be adjusted to be transparent? In Fig. 22f, the arrow is too large. The tail of the arrow also does not face area

6.      The conclusion is better written in sections, so it is more clear.

Author Response

Thank you very much for pointing out important comments to the article. I have tried to include most of them. I have introduced changes to the manuscript and send it via the journal's IT system. Here are my answers and explanations of the relevant points from the review:

Ad. 1. The obtained photos of the structures are very different from each other due to the non-uniform illumination of the photographed surface. For this reason, I decided to include them, but in black and white. It would be difficult to standardize the photos and include them in the article in a reasonable time. I hope this will not significantly worsen the image effect.

Ad. 2. I have included an appropriate explanation under the table no. 2.

Ad. 3. Agree that it would be better to include labels (a) (b) (c) (d) in the right corner of the pictures. However, I tried to adapt to the tagging system that the publisher prefers. I hope it won't be a big problem.

Ad. 4. The photos in Fig. 22c come from a similar area but are taken at different magnifications. The intention of presenting them together in one drawing was to show the scale of the studied area. In the upper figure, I have marked the area of the target structure and hardness tests. I explained the meaning of the small arrows in the caption under the drawing. The drawings with selected SEM images illustrate the tests carried out in a specific area marked with numbers from 1 to 8, which should be associated with Figure 22c where the hardness values are given. If it is an unclear way of presenting the results, maybe the hardness values for each area could be given in the caption of each drawing with SEM tests.

Ad. 5. The background of the numbered field in Figures 22d-22f has been selected so as to unambiguously relate the SEM test result to the appropriate numbered area. It would be possible to change the convention, but in my opinion, it would not improve the identification process. If necessary, I will of course make a change. The arrow in fig. 22f has been corrected.

The conclusions have been redrafted as suggested.

Yours faithfully

Reviewer 3 Report

Present manuscript is devoted to analysis of impact of acoustic vibrations on the laser beam remelting process. The research is interesting and also shows interesting finding. However, reviewer have some serious concern regarding manuscript and also some suggestion for improvement of manuscript.

1.      The current research is based on earlier published article “A Novel Method of Supporting the Laser Welding Process with 326 Mechanical Acoustic Vibrations” by authors and it has also been cited multiple times in the manuscript. However, it should be mentioned clearly in abstract or introduction section for reader.

2.      Basic discussion regarding the finding of earlier research finding can be included in introduction or in materials and method section.

3.      Unit should be mentioned in Figure 1 and also in other Figures

4.      The basic data received in the earlier experiment should be mention or it is creating confusion. E.g. sample 3 mentioned in Result section. Sample no 3 has been mentioned in multiple place. However, the details of sample 3 is missing in the manuscript.

5.      Caption of the Figure should be standardize. The caption mentioned in the Figure 20 “Metallography a) and SEM b) of the sample no 1 reached with 50 Hz acoustic vibration delivered transversely” creating confusion regarding Figure 20 (a) and (b). It seems that authors want to write a) Metallography and b) SEM. Similar non-standard use of caption has been observed with other figures.

6.      Figure 20, 21, 22 shows the SEM pictures, it metallurgical or microstructural changes could be marked or mentioned in the figure.

7.      Figure 21 refer the sample no 2. The details of sample no 2 should be mentioned.

8.      In discussion section the word “welds” has been mentioned where as remelting term has been used though out the manuscript. Use of uniform word may avoid confusion.

9.       In discussion section (page 19) discuss regarding the change in the chemical composition after laser remelting subjected to vibration. However, the chemical or elemental concentration changes also depend of multiple factors like laser power, cooling rate, and heating rate etc.  Authors could refer earlier published articles to improve the discussion.

10.   In discussion section should answer the question of “How vibration helps in change the structure, which also affect the hardness ?”

11.   The writing in the manuscript need to be improved. The sentences need to be simplified. Authors may look into following sentences for improvement:

a.      “There 37 you can find the results of structure studies after remelting with a CO2 laser, hardness 38 distributions and analysis of the shape of the laser impact areas, especially in [5].” Page 1, line 37-39

b.      “The results of the analysis carried out are therefore a continuation of the study [5] and constitute a comprehensive methodological and substantive whole  with this work.” Page 2, line 51-53

c.      In the experimental tests, the previously obtained results of tests [5] were used, carried out on a pipe made of P235GH steel with a diameter of 60 mm, a wall thickness  of 3.2 mm and a length of 500 mm.”  Page 2, Lines: 55-56.

d.      “While the heat flow during welding can be treated as a constant and continuous  phenomenon, the introduction of mechanical vibrations to the base material is not.”  Page 3, line 75-76.

e.      The word sound energy/ pressure is creating confusion “The  nature and form of the vibrations, the introduced sound energy / pressure, and the direction and environmental conditions inside and outside the pipe there are important here.” Page 32, line 76-78.

f.       Figure 22 should contain the unit and also the description of annotation in the figure.

Author Response

Thank you very much for your critical evaluation of our manuscript. Here are responses to comments and suggestions:

Ad. 1. As suggested, appropriate explanations are provided in the chapters “Abstract”, “Introduction” and “Materials and methods”.

Ad. 2. Suitable explanations of the suggested chapters are provided.

Ad. 3. Necessary units are provided in figure captions.

Ad. 4. The description of the conditions for obtaining sample No. 3 has been supplemented in appropriate table no 1.

Ad. 5. Captions under the figures have been standardized as suggested.

Ad. 6. Appropriate captions under the figures have been added.

Ad. 7. All parameters relating to the tested samples are listed in Table 1.

Ad. 8. Terminology was standardized as suggested.

Ad. 9. References to previous studies from item [5] were included in the discussion.

Ad. 10. The discussion includes a sentence on how acoustic vibrations affect the structure and hardness of the obtained remelts.

Ad. 11. All remarks regarding linguistic shortcomings have been considered in the appropriate places in the text.

Yours faithfully

Round 2

Reviewer 1 Report

The authors have succeeded to answer most of made suggestions / requests.

The only minor changes I am suggesting before paper could be considered accepted for publication would be:

1. re-considering of counting of images in the text as following"

- Fig. 23 a and b could be kept as they are now:

- Fig. 23 c - I personally would consider as being Fig.24

- Fig.23 d, e and f - I personally would consider it as being Fig. 25 a, b and c

- Fig.24 a and b - would be Fig. 26 a and b in the new variant

- Fig. 25 a and b - would be Fig. 27 a and b in the new variant

- Fig. 26 would become Fig 28 in the new variant

2. I personally would completely rephrase the last paragaraph presented at section 5. Conclusions as follows (see Lines 641-646):

"The result of the caried out analyzes carried out here shows that it is possible to construct a modified laser head for welding processes involving mechanical vibrations.  However, to thoroughly answer the question of how acoustic vibrations affect the welding process, a numerical superposition of thermal phenomena and mechanical vibrations will be needed in the future". 

My suggestion is to go like this and completely remove finally the last sentence "Conceptual work on such a solution is already underwayin the process" since details about it are not provided in the text finally. From the context it is possible to deduce that such solution is aimed to be study / analyzed / conceived in the future.

With these last two made observations / last minor changes requested solved by the authors, I agree with the publishing of this paper in Materials MDPI journal. The changes that the authors have made to the manuscript makes much better the paper to be understood now and also the comparison between the initial results published in the previous paper and the new ones published in the current paper is looking much clear now.

Author Response

Thank you very much for your careful evaluation of the manuscript. We took into account all suggestions regarding the numbering of figures and the change of the last paragraph in the conclusions. We believe that now the amended article has gained a lot as a whole.
  Kind regards

Reviewer 3 Report

The authors have done significant improvement in the manuscript. Authors could correct minor typographical error, in final manuscript editing. 

Author Response

Thank you very much for the substantive evaluation of the manuscript.
  Kind regards,